# Repellent Activity of Clove Essential Oil Volatiles and Development of Nanofiber-Based Dispensers against Pear Psyllids (Hemiptera: Psyllidae)

**DOI:** 10.3390/insects13080743

**Published:** 2022-08-18

**Authors:** Bruna Czarnobai De Jorge, Hans E. Hummel, Jürgen Gross

**Affiliations:** 1Institute for Plant Protection in Fruit Crops and Viticulture, Julius Kühn-Institut, Federal Research Centre for Cultivated Plants, Schwabenheimer Str. 101, 69221 Dossenheim, Germany; 2Plant Chemical Ecology, Technical University of Darmstadt, Schnittspahnstr. 4, 64287 Darmstadt, Germany; 3Organic Agriculture, Justus-Liebig University of Giessen, Karl-Gloeckner-Str. 21C, 35394 Giessen, Germany; 4Illinois Natural History Survey, Biodiversity and Ecological Entomology, Champaign, IL 61820, USA

**Keywords:** repellent, hemiptera, psyllidae, traps, integrated pest management, essential oil, nanofibers

## Abstract

**Simple Summary:**

Pear psyllids are responsible for transmitting the pathogen ‘*Candidatus* Phytoplasma pyri’, causing the pear decline disease. Specific repellents are a potential method for controlling this pest and reducing the spreading of phytoplasma pear in orchards. Little is known about the chemical ecology of these insects. Based on recent literature, we tested the behavioral reactions of *Cacopsylla pyri* and *Cacopsylla pyricola* to the major synthetic compounds of clove essential oil in olfactometer experiments. In addition, we developed new nanofiber-based repellents dispensers and evaluated their repellent and host-odor masking activity in the laboratory and later in the field. Results demonstrate that only the synthetic mixture of the three major compounds of the clove essential was repellent to both psyllids species. When volatiles were formulated in nanofibers, they were not only repellent but could mask the odors of pear plants, disrupting the insects from finding the host. In the field, no differences in psyllids captures were observed in color-attractive sticky traps with repellent-loaded nanofibers or nanofibers without repellent. Our study also evaluated the release rates of volatiles from the nanoformulation. We discussed using nanofibers as volatile dispensers and the improvements necessary to use repellents as a management tool for pear psyllids in the field.

**Abstract:**

Pear psyllids are the main vectors of the pathogen ‘*Candidatus* Phytoplasma pyri’ causing pear decline. Based on earlier reports, we tested the behavioral activity of the major synthetic compounds of clove essential oil (eugenol, eugenyl acetate, and β-caryophyllene) against *Cacopsylla pyri* and *C. pyricola*. Of six mixtures tested in olfactometer assays, a formulation consisting of three specific compounds (M6 mixture) demonstrated a repellent effect on both psyllid species. In addition, this formulation masked the odor of the host *Pyrus communis* cv. Williams Christ, disturbing the host finding ability of *C. pyri*. Electrospun fibers were produced with biocompatible polymers poly(ε-caprolactone), cellulose acetate, and solvents formic acid and acetic acid, loaded with the repellent mixture to test their efficacy as dispensers of repellents in laboratory and field. The fibers produced were repellent to *C. pyri* and effectively masked the odors of pear plants in olfactometer tests. In a pear orchard, we compared the captures of pear psyllids in green-colored attractive traps treated with nanofibers loaded with M6 mixture or unloaded nanofibers (blank). The result showed no differences in the captures of *C. pyri* between treatments. The release rates of volatiles from the fibers were evaluated weekly over 56 days. The fibers were able to entrap the major compound of the M6 mixture, eugenol, but the release rates were significantly reduced after 21 days. Our results suggest that biodegradable dispensers could be produced with electrospinning, but further improvements are necessary to use repellents as a management tool for pear psyllids in the field.

## 1. Introduction

Pear decline (PD) is the disease caused by ‘*Candidatus* Phytoplasma pyri’, a phytoplasma that belongs to the Apple proliferation group (16SrX) [1]; it is a worldwide disease that affects pear production by causing severe damage and losses in this crop [2]. The transmission and spread of this disease between orchards are accomplished by pear psyllids (Hemiptera: Psyllidae: *Cacopsylla pyricola*, *C. pyrisuga*, *C. pyri*) [3]. In North America and England, the primary known vector is *C. pyricola* (Foerster), but in parts of Europe, *C. pyri* (L.) has been identified as the main vector [4,5]. Transmission of PD by *C. pyri* has already been demonstrated in Italy [6] and France [7], suggesting that this psyllid is probably the most important vector in the Mediterranean area. Although transmission capability has not yet been evaluated, *C. pyri* is also the most common psyllid in pear orchards in Spain [5].

Most phytophagous insects are specialists [8], who use volatile cues to find and accept their host plants [9]. When locating and selecting a host, pear psyllid’s behavior, visual cues, such as plant color, and olfactory cues, such as plant volatiles, play critical roles. In the visible spectrum, specific wavelengths of light, notably those corresponding to the color green, are more attractive to *C. pyri*, influencing movement and host choice [10]. Pest management practices have taken advantage of this behavioral aspect by using green sticky traps to monitor the *C. pyri* population [10]. Behavioral studies with pear psyllids suggest that chemical cues within the plant or surface affect host selection, feeding, and oviposition [11,12]. Past research on chemically mediated multitrophic interactions between host plants and insect vectors reveals the possibility of identifying infochemicals influencing insect vectors’ behavior to be used in biotechnical pest control. For example, repellent compounds or masking odors (also called attraction inhibition) can either reduce the attractiveness of the host or disrupt the host-seeking behavior by the odor cue [13]; however, the relationship between non-host plants and repellency is unclear. Although some insect herbivores are repelled by non-host odors [14,15], there is still limited understanding of how interspecifc variation of plant odors influences host-seeking and acceptance of specialist herbivores. As reviewed by Deletre et al. (2016) [13], a number of different mechanisms could cause repellency: True repellents (also expellent), contact irritancy (also called landing inhibition), deterrence (also called antifeeding) and odor masking (also attraction inhibition). Non-host odors can mask the recognition of host odors by physically disrupting the reception of host compounds, effectively cancelling the intended effect.

Biopesticides encompass a large number of technologies, from microbial organisms to botanicals. Among the botanicals, essential oils are a significant category that began to develop with research in the 1980s [16]; they are derived from aromatic plants that developed myriad constitutive and induced chemical defenses against herbivorous insects [17]. Representative lists of essential oils have been published by Bakkali et al. (2008) [18] and Sosa & Tonn (2008) [19]. Most publications document the immediate effects (acute toxicity or repellency) of given essential oils on several arthropod taxa, frequently based on assays lasting less than 48 h. Papers with similar approaches demonstrated effects across a wide range of taxa or species. Although many essential oils have proven successful against horticultural pests, only a few are used in agriculture [20]. There not many studies on the repellent effect of plant essential oil and other compounds on psyllids of agricultural importance. These natural products have the potential to provide efficient repellency safe for humans, the environment [21], and beneficial insects [22]. The isolation and identification of attractive and repellent info chemicals may also lead to innovative control strategies [23].

Clove essential oil has been widely studied for its insecticidal and repellent activities against many species of pests [24,25,26], such as fire ants [27], aphids [28,29], weevils [30,31], moths [32,33], and psyllids [14,34]. Five constituents were detected in the essential oils by GC-MS, accounting for 99.89% of total content, and the major constituents were eugenol (88.61%), eugenol acetate (8.89%), and β-caryophyllene (1.89%) [34]. In the study, they evaluated the acute toxicity of clove essential oil against *C. chinensis* and observed that, in the field, clove essential oil treatment could reduce the number of nymphs in a concentration-dependent manner; however, there is no report on the possible repellent activity against pear psyllids (*C. pyri* and *C. pyricola*). Since clove is a less related plant to pear and has less similar odors, we predicted that they are more likely to repel *C. pyri* and *C. pyricola.* The main component of clove essential oil is notably eugenol; however, there are differences in the composition of its constituents; those differences may be caused by the variation of vegetative state, growing season, and the places of origin [35,36,37], making it challenging to evaluate biological relevant effects on insects’ behavior. Plant-based repellents can be compared to synthetics; however, essential oil repellents tend to be short-lived in their effectiveness, depending on their volatility [21].

Nanotextiles are emerging as the most potent tool for future sustainable agricultural production advances. The innovative application of nanotechnology, specifically polymeric nanofibers as vehicles for dispensing semiochemicals, was reviewed by Czarnobai De Jorge and Gross (2021) [38]. These formulation techniques allow active ingredients’ slow and persistent release and reduce the required amount by minimizing their waste. The notoriety of the nanoscale delivery system in agriculture is due to its improved stability against degradation in the environment. Nanoformulations such as pheromones dispensers have proven to be efficient for prolonged periods in laboratory experiments [39].

Therefore, this study aims to evaluate the effect of clove essential oil major constituents against *C. pyri* and *C. pyricola* and to test if the formulation and incorporation of synthetic volatiles of clove in nanofibers can be efficiently applied repellents for pear psyllids in the field. The release rates of the volatiles from nanofibers were also determined.

## 2. Materials and Methods

### 2.1. Insects

Adults of *C. pyri* used for olfactometer trails were sampled daily from *Pyrus communis* cv. Williams-Christ trees at the Julius Kühn-Institut (JKI) experimental orchard in Dossenheim, Germany. Insects were captured using the beating tray method, according to Weintraub and Gross (2013) [23], and the species were identified using the key of Burckhardt and Hodkinson (1986) and Ossiannilsson (1992) [40,41]. *C. pyricola* individuals were obtained from a laboratory culture maintained at the Julius Kühn-Institut (JKI) (Dossenheim, Germany); they were maintained without exposure to insecticides on healthy potted *Pyrus communis* cv. Williams Christ plants in a 47.5 × 47.5 × 93 cm rearing cages (Bug Dorm, NHBS, Devon, UK) in a climatic chamber with 20 °C day and 15 °C night temperatures under long-day conditions (L16:D8) and 55% relative humidity [42].

### 2.2. Materials

#### 2.2.1. Clove Oil Components

All volatiles examined in this study were selected from the literature [34]. The compounds used in the experiment were eugenol (99% purity), eugenyl acetate (98% purity), β-caryophyllene (80% purity) and were purchased by Sigma-Aldrich GmbH, Taufkirchen, Deutschland.

#### 2.2.2. Preparation of Polymer Solution

Poly-(ε-caprolacton) (PCL, Mw 80,000 g/mol, Sigma Aldrich, Burlington, MA, USA), cellulose acetate (CA, Mw 50,000 g/mol, Carl Roth GmbH, Karlsruhe, Germany), acetic acid (ROTIPURAN^®^ ≥ 99% purity, LC-MS Grade, Carl Roth), formic acid (ROTIPURAN^®^ ≥ 98% purity, p.a., ACS, Carl Roth). PCL/CA (1:1) were dissolved in acetic acid: formic acid mix (1:1). The concentration was kept at 15% (*w*/*v*) with respect to solvent. The solutions were made by continuously stirring the mixture overnight at 1500 rpm with a magnetic stirrer at room temperature to obtain a completely homogenous solution. The concentration of the synthetic mixture in the solutions was 5% and 10% (*v*/*v*).

### 2.3. Electrospinning

We used a commercially available electrospinning unit from Linari Engineering, Italy. A 4 mL polymer solution was taken from the stock via a 5 mL syringe fitted with a nozzle of 0.8 mm diameter. The syringe was placed on an automated syringe pump that provided a controlled flow rate of 0.8 mL/h. Fibers were collected on aluminium foils placed on the grounded collector at an 11 cm distance from the needle tip (nozzle-to collector distance) and an applied voltage of 12 kV. The resulting fibers were kept under a ventilated hood overnight for complete solvent evaporation and stored at −20 °C until performing the tests.

### 2.4. Olfactometer Assays

To determine female pear psyllid’s response to synthetic volatiles contained in clove essential oil, olfactometer tests were performed. A dynamic glass Y-shaped olfactometer (12.5 cm stem length, 8 cm arm’s length, 1 cm inner diameter, and an internal angle between arms of 75°) was mounted on a board at an angle of 40° from the horizontal plane. The experiments were conducted as previously reported [43], to investigate the responses of *C. pruni* to host odors. The Y-olfactometer was placed in a dark room and was illuminated with a LED magnifying light of 280 lx (Purelite, UK) 45 cm above the middle of the olfactometer. All tests were performed between 12:00 a.m; moreover 6:00 p.m. at room temperature and humidity (20–26 °C and 30–35% RH). On cloudy and rainy days, no tests were performed, since psyllid motivation is lower [43]. The air entering the system (flow of 40 ± 1 mL/min) passed through a charcoal filter and was humidified, then pumped through the odor source into one of the olfactometer arms. The detailed construction of the bioassay procedures is described elsewhere [43,44]. The preferences of field-collected *C. pyri* females and *C. pyricola* females from lab culture were tested. Single psyllids were collected in 1.5 mL plastic vials and left to starve overnight in the refrigerator kept at 6 °C. About an hour before the experiment, insects were transferred to room temperature. A single female was introduced at the base of the Y-tube olfactometer and observed for 5 min. Only the psyllids were counted that entered one of the test arms and passed 2/3 of the tube, remaining there for at least 30 s. Individuals that did not make a choice within 5 min were recorded as “not responding.” After each experiment, all olfactometer parts were rinsed with ethanol (70%) and heated at 230 °C for three hours with the exception of the plastic valves (60 °C for 2 h). After five repetitions, the side on which the treatment was presented was swapped to avoid positional bias. A minimum of 26 individuals were tested in each trial. The following experiments were conducted:(a)Three substances and mixtures (Table 1) were used to study the repellent effect of synthetic compounds contained in clove essential oil on female adults of *C. pyri* and *C. pyricola*. A stock solution of all mixtures (concentrations of each compound, Table 1) was prepared and then diluted with methanol to 5% final concentration. For each substance or mixture, 3 µL were applied onto a filter paper (2.5 × 2.5 cm) and kept for five minutes under a ventilated hood for solvent evaporation. The filter paper and the volatiles were presented in one arm of the olfactometer simultaneously with purified air in the other arm. The volatile samples were renewed on an hourly basis.(b)To observe the host odor masking effect of the M6 mixture of clove volatiles in the presence of pear tree volatiles, one potted pear plant cv. ‘Williams-Christ’ (50 cm) was carefully wrapped in oven plastic bags (Toppits, Melitta, Minden, Germany, 31 × 50 cm) and connected to one of the test arms. In the other arm an empty oven bag was used as control. Air was pumped to the wrapped plants and empty oven bag for 30 min before the experiment to stabilize the system. The airflow passing at the outlet of the oven bags and entering each olfactometer arm was controlled by a flowmeter (MASS-STREAM, M + W Instruments, Allershausen, Germany) and adjusted with plastic valves. In the olfactometer arm contained the pear volatiles, a filter paper containing the M6 mixture, as described above, was added, and the other arm was supplied only with solvent.(c)For the test with volatiles encapsulated into nanofibers, a 3 mg piece of nanofibers loaded with 5% of the volatile M6 mix was offered in one olfactometer arm. As a control, purified air was pumped into the other arm. The masking effect of the nanoformulation was also tested as above: in one olfactometer arm, nanofiber + M6 was offered with the pear volatiles, and the other arm was supplied only with air.

### 2.5. Field Trials

Field experiments were conducted to observe if attractive stick traps equipped with nanofiber formulated with the repellent mixture (M6) would reduce the catches of psyllids in comparison with traps equipped with blank nanofiber. The test was performed in a non-commercial pear orchard in Germany (Julius Kühn-Institute, Dossenheim, Germany). The traps were fabricated from transparent 0.02-mm-thick rigid vinyl plastic cylinders, 9 cm in diameter × 25 cm in length, coated with a thin layer of insect glue, provided by Insect Services GmbH, Berlin. Traps were designed with 16 holes (1.5 cm diameter) evenly distributed (5 cm distance between each other) (Figure 1a). Test traps were equipped with green-colored transparent films (#068) as described by Czarnobai De Jorge et al., 2022 [10] for the attraction of pear psyllids. Two treatments were tested: traps equipped with 3 g of nanofibers loaded with 10% M6 (n = 5) or equipped with blank (empty) nanofibers (n = 5). Fibers were folded and inserted in voile bags (7 × 7 cm) tight with a cord and hung inside the traps (Figure 1b,c). We designated five blocks of pear trees for nanofiber assessments. Within each block, traps were at least 20 m separated from each other. Traps were hung at 1.5 m height and deployed between two pear trees.

Traps were evaluated weekly for six weeks between 21 June 2021 and 2 August 2021. Captured psyllids were identified, counted, and removed from the traps.

### 2.6. Release Rates Study

#### 2.6.1. Micro-Chamber/Thermal Extractor Method

A six-chamber unit with the capability to analyze six samples at once was employed in the experiment (Micro-Chamber/Thermal Extractor™ (µ-CTE™), Markes International GmbH, Offenbach, Germany). Nanofiber samples containing 10% M6 were cut in small pieces, each piece 30 ± 0.1 mg and then placed into the individual chambers made from stainless steel (n = 6); they were incubated at an optimal temperature of 24 °C and purged with 100 ± 0.1 mL/min constant flow with high-purity synthetic air (Linde, Munich, Germany). Following an equilibration period, clean sorbent tubes (Tenax TA, Markes, 35/60) were connected to the outlet of each individual micro-chamber to collect the volatiles purged from the nanofibers sample. Volatiles were collected for one minute. After sampling, the sorbent tubes were sealed tightly by Swagelok caps for storage. Immediately before the tests n-dodecane (50 ng) was added in the sorbent tubes as an internal standard. Subsequently, tubes were loaded onto an automated thermal desorption prior to analysis by GC-MS (details see below). To observe the release rates of the M6 VOCs from nanofibers, new fibers were left under a laboratory hood after the sampling start and sampled every seven days for 56 days, as described above.

#### 2.6.2. Thermodesorption—GC/MS

An automated thermal desorption system (TurboMatrix™ ATD 650, PerkinElmer, Rodgau, Germany) connected to a gas chromatograph coupled with a mass spectrometer (GC–MS) [43] was used to analyze M6 volatiles released from nanofibers. The methods and equipment used were the same as that described by Gallinger et al., 2020 [43]. Sample tubes were thermally desorbed for 10 min at 250 °C. The temperature of the cold trap (Tenax TA) was held at −20 °C during the desorption process, then heated at 99 K/s to 250 °C and desorbed for 1 min. The chromatographic separation of the compounds was performed using a PerkinElmer Clarus R 680 GC with a nonpolar Elite-5 capillary column (30 m × 0.25 mm, film thickness 0.25 μm, PerkinElmer). The GC initial oven temperature was held at 40 °C for 1 min; subsequently increased to 180 °C at 5 K/min, finally to 280 °C at 20 K/min and held for 6 min. The quadrupole mass detector was operated as follows: the electron impact (EI) mode was set to 70 eV, the GC inlet line temperature was 250 °C, and the ion source temperature was 180 °C.; full-scan mass spectra within the range of 35–350 *m*/*z*.

#### 2.6.3. Identification and Quantification of Essential Oil Compounds with AMDIS

Evaluation of GC–MS chromatograms was carried out using “Automated Mass spectral Deconvolution and Identification System” (AMDIS, V. 2.71; National Institute of Standards and Technology NIST, Boulder, CO, USA) following the protocol in Gross et al. (2019) [45]. The compounds in the samples were identified by comparing retention times, RI and mass spectra with available standard compounds, according to Weintraub and Gross (2013) [23]. As identification criteria a minimum match factor of 80% was set and the relative retention index deviation of 5% from reference values was set as limit; level: strong; maximum penalty: 20; and “no RI in the library”: 20. For quantification, the peak areas were integrated after deconvolution with AMDIS. The settings for deconvolution were: component width: 32; adjacent peak subtraction: one; resolution: low; sensitivity: medium; shape requirements: low; signal-to-noise: ≥50 [43]. The relative content of each compound in the samples was calculated by comparison with the peak area of 50 ng n-dodecane (internal standard).

### 2.7. Statistical Analysis

The data evaluation was conducted with the program R (Version 1.2.5033). All figures were generated using the “ggplot2” package. Binomial tests were used to determine the significance of choice between treatments odor vs. control in two-choice Y-olfactometer assays. Non-responder were excluded from this analysis. For field trials, the averages of psyllids collected by trap were compared using generalized linear models (GLMs), assuming a quasi-Poisson distribution (count data with overdispersion). We considered “week” and “trap type” as a fixed factors in the model. Using a deviation analysis (F-test, link function: “log”), we investigated whether the factor ‘trap’ significantly influenced the number of insects oriented towards the traps. Interactions between the temporal repetitions of experiments and trap type “week: trap” were included in the model as fixed factors. The evaluations of the release rate of volatiles from nanofibers were performed with GLM quasi-Poisson model, considering “days” of exposure of nanofibers as a fixed factor. The data was log-transformed for the analysis. The deviation analysis was performed to evaluate the impact of exposure time “days” on the release rates (F-test, link function: “log”). Fitted models were tested for the field experiment, and nanofibers release rates with a residual diagnostic for hierarchical (multi-level/mixed) regression models. Multiple pairwise comparisons were calculated with estimated marginal means and 95% confidence intervals with the function emmeans from the “emmeans” package [46] and *p* values adjustment by Tukey.

## 3. Results

### 3.1. Olfactometer Assays

Female *C. pyricola* were significantly repelled by M5 and M6 (Figure 2, binominal test, *p* = 0.035 and *p* = 0.017, respectively); however, the other blends and single compounds vs. methanol did not affect their behavior (Figure 2). Females of *C. pyri* were significantly repelled by M1, M4 and M6 (Figure 3, binominal test, *p* = 0.03469; *p* = 0.00151; *p* = 0.04329, respectively). In this case, β-caryophyllene and eugenyl acetate alone were not repellent to *C. pyri* females (Figure 3). The total motivation level (percentage of psyllids that made a choice) was higher in *C. pyri* than in *C. pyricola* (80% and 68%). We observed that M6 was effective in repelling both psyllid species in choice assays (Figure 2 and Figure 3).

*C. pyri* chose more frequently the untreated arm of the y-tube olfactometer in both experiments where the pear volatiles were present (significant *p*-values from binomial tests; Figure 4a,c). Clearly, the M6 and the nanofibers with M6 were effective in masking pear tree odors and disrupting host findings by *C. pyri* females in olfactometer tests; moreover, when no pear odors were present, a higher significant difference was observed (Figure 4b; *p* < 0.0001).

### 3.2. Field Trials

The model showed no differences in psyllids catches between the two different trap types in the field trials (F_1,53_ = 0.83; *p* = 0.34). Statistically, *C. pyri* catches on traps equipped with nanofibers and M6, and the blank ones were similar; moreover, the interaction between trap type capture (blank and M6) and week (F_5,48_ = 0.25; *p* = 0.93) was not significant. According to the model, the only significant factor was week (F_5,54_ = 10.71; *p* = 0.0001). The number of *C. pyri* captured in traps was significantly influenced over time (Figure 5). The mean number of captured *C. pyri* per week was 26.80 ± 2.30 in Green + Blank and 29.53 ± 3.44 in Green + M6 traps (Figure 6). *C. pyri* was the most abundant species in the orchard used for the field trials, and only a few *C. pyricola* individuals were captured. For that reason, all results presented are related to *C. pyri*.

### 3.3. Release Rates

No traces of the solvents acetic acid and formic acid used to produce the fibers were detected in the chromatograms. The compound β-caryophyllene was not detected in the fibers, and eugenyl acetate was only detected during the first two weeks of tests. Since eugenol was consistently detected in all the samples, the statistical analysis of the released rates was performed based on this compound. The model could demonstrate that release rates of eugenol from nanofibers were altered over time (F_5,58_ = 19.34; *p* < 0.0001). There was no difference between the amount of released eugenol from new nanofibers and after seven and 14 days of exposure (*p* = 0.99 and *p* = 0.66, respectively; Figure 7). After 21 days, the amount of eugenol released from nanofibers was statistically similar to fibers that were exposed for seven and 14 days (*p* = 0.34 and *p* = 0.78, respectively), but already reduced in relation to the rates of new nanofibers (*p* = 0.027). The release rates of eugenol decreased after 28 d and remained constant until the end of the tests (Figure 7). The average release rate of eugenol from the nanofiber dispenser was calculated to be 0.3 ± 0.06 mg/day.

## 4. Discussion

Like many other essential oils, clove oil is widely used as herbal medicine and spice; they are generally recognized as safe for human health [26]; however, their composition may vary considerably between plant varieties, vegetative state, growing season, and within the same variety from different geographic areas. Even though the constituents in clove bud oils can differ, there is no doubt that eugenol is the principal constituent in clove oil in all growing areas [24]. Sigma-Aldrich Chemical Co., Ltd. (Poole, England) indicates that the oil mainly contains eugenol (78.00%) and β-caryophyllene (13.00%) [47]. The difference may be caused by variations in the vegetative state, growing season, and the places of origin [35,36,37]. Constituents of clove bud oils from India and Madagascar differ significantly concerning eugenol (70.00 and 82.60%) and β-caryophyllene (19.50 and 7.20%), and eugenyl acetate (2.10 and 6.00%) [36]. Although the repellent activity of essential oil is generally attributed to some particular compounds, a phenomenon among these metabolites may result in higher bioactivity than the isolated components [48,49]. Omolo et al. (2004) [50] compared repellent activity between the essential oil and synthetic blends formulated with their major constituents. For some of these formulated blends, the repellency was much smaller than the corresponding essential oil.

Our study demonstrated that a synthetic formulation of clove essential oil could be effective as a repellent to *C. pyri* and *C. pyricola* females. We studied psyllid behavior in olfactometer choice tests and in the field. By conducting choice assays, we evaluated the relative preference or avoidance between single compounds and mixtures of clove essential oils for their ability to repel *C. pyri* and *C. pyricola*, and in altering host-seeking behavior in *C. pyri.* Experiments with this configuration which assess the number of insects avoiding treated plants in the olfactometer may be more predictive of insect behavior in the field. Furthermore, olfactometer assays separate the insect from contacting the odor source, specifically testing host-seeking behavior [51,52]. Here, the M6 mixture was the only one that presented a repellent effect on both psyllid species (Figure 2 and Figure 3). Insect repellent is defined as a phenomenon that prevents a pest from tracking, locating and/or recognizing its host [13]. A true repellency and a host-odor masking effect could be demonstrated in lab bioassays; however, there are no clear guidelines for repellent field tests for agricultural pests since there are few examples of products used for this purpose and their effectiveness [20]. For the field trial, we produced nanofibers with a higher concentration of M6 (10%) to enhance and extend their effectiveness in field conditions; however, in the field, we observed no difference in *C. pyri* captures on traps equipped with M6 nanofibers with and without treatment (Figure 6); this result can indicate that the cause of the treatment failure may be attributable to the distribution of the dispensers in the area rather than an ineffective repellent formulation. Perhaps the outcomes of the field trial could be different if a larger area of the orchard were treated with the repellent formulation rather than a trap point source. In this scenario, a substantial inhibitory effect on the pear psyllids population would be expected with a significant reduction in psyllids trapping. Such modifications in the experiment design could be performed for future investigations.

Color is a factor that may play a more significant role in the psyllid’s behavior as VOCs on host finding. The repellent effect of eugenol could be counteracted by the presence of colored traps. Some psyllids’ and aphids’ chemotactic responses are enhanced when visual cues are present [53,54,55]. In our olfactometer bioassays visual cues have been absent. Olfactory cues may guide psyllids along specific odor profiles, but once a visual cue is added, it could be stronger than the olfactory one for guidance.

Nanofibers encapsulated the major compound of M6, eugenol; this was proven by the evaluation of release rates from new nanofibers, since eugenol was detected for 56 days (Figure 7). The second compound in the M6 mixture, eugenyl acetate, was only detected in new fibers, and after 7 and 14 days of exposition in trace amounts, as it was rapidly released from the fibrous polymeric structure. β-caryophyllene was not detected in any nanofibers samples, demonstrating that the encapsulation of this component was not successful in these polymers, and it was probably already lost during the electrospinning process. The reason, e.g., oxidization, should be subject to further studies. Even though eugenol has been demonstrated to be a suitable repellent to *C. pyri* in olfactometer trials, the insect’s behavior was not disrupted in the open field, and it did not prevent the insect from finding and being caught in the trap. The olfactometer results suggest that eugenyl acetate has a synergistic effect with eugenol and β-caryophyllene in repelling of *C. pyricola*. Since the produced fibers did not contain these compounds, they would not be as effective against this species in the field. β-caryophyllene has already been described as a repellent against *Diaphorina citri* [56,57]. The repellent effect of β-caryophyllene together with eugenyl acetate against the pear psyllid *C. pyri* has been demonstrated in our tests, but not as a single compound; however, it could probably have a repellent effect in a dose-dependent matter.

This research demonstrates that nanofibers could be produced with biocompatible, biodegradable, non-toxic, and environmentally friendly polymers PCL/CA and solvents acetic acid and formic acid as described elsewhere [58,59,60,61,62]; this is an excellent advance since biopolymers and biodegradable polymers that could be easily electrospun may be applicable in the protection against pests in the future, mainly if industrial electrospinning instruments are developed to produce nanofibers on a larger scale [38].

Further tests with higher amounts of the three active components of clove essential oil should be performed, as well as different distribution and a higher number of nanofiber materials in a given area to test the efficacy of the repellent in the field. Different fiber formulations with other polymer compositions could also be produced to stabilize the release rates of those compounds and enable the use of repellents to manage psyllids in the field. Our future goal is to develop an environmentally sound and cheap repellent emitting device with minimal side effects on non-target insect species in the orchard.

## Figures and Tables

**Figure 1 insects-13-00743-f001:**
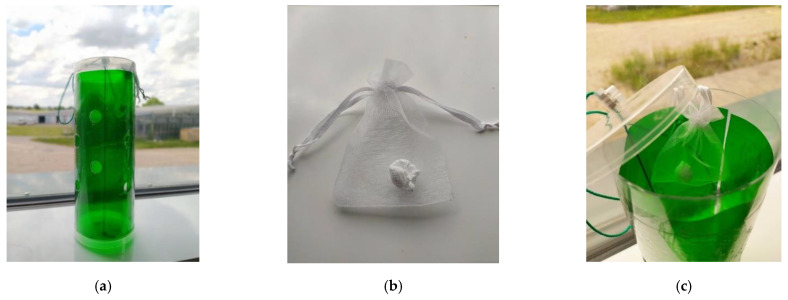
Green cylindric sticky trap used in this study (**a**); Nanofiber material inside a voile bag (**b**); voile bag enclosed in the trap (**c**).

**Figure 2 insects-13-00743-f002:**
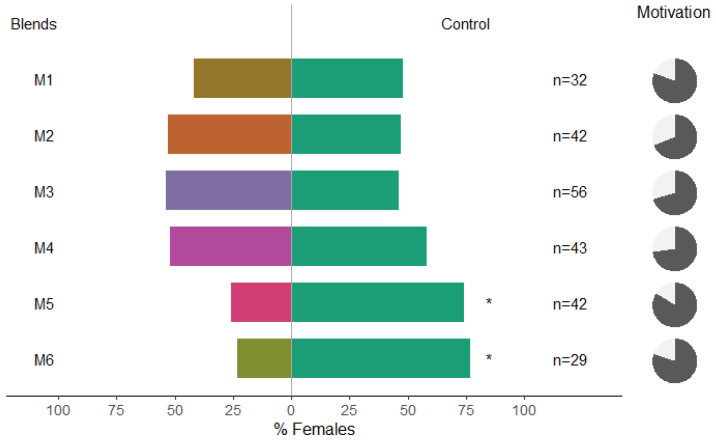
Results of Y-olfactometer-bioassays testing synthetic clove VOCs dissolved in methanol with adults of *Cacopsylla pyricola*. Pure methanol was used as control. The letter “n” corresponds to the numbers of insects used for the tests. Results are shown as the percentage of psyllids in each arm. Statistically significant differences are indicated with * (*p* < 0.05). The percentage of psyllids that made a choice (dark gray) or not (light gray) is presented as pie charts on the right.

**Figure 3 insects-13-00743-f003:**
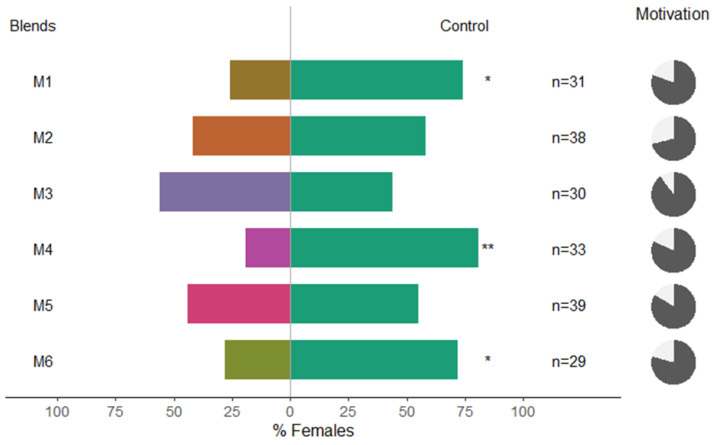
Results of Y-olfactometer-bioassays testing synthetic clove VOCs dissolved in methanol with adults of *Cacopsylla pyri*. Pure methanol was used as control. The letter “n” corresponds to the numbers of insects used for the tests. Results are shown as the percentage of psyllids in each arm. Statistically significant differences are indicated with * (*p* < 0.05); ** (*p* < 0.01). The percentage of psyllids that made a choice (dark gray) or not (light gray) is presented as pie charts on the right.

**Figure 4 insects-13-00743-f004:**
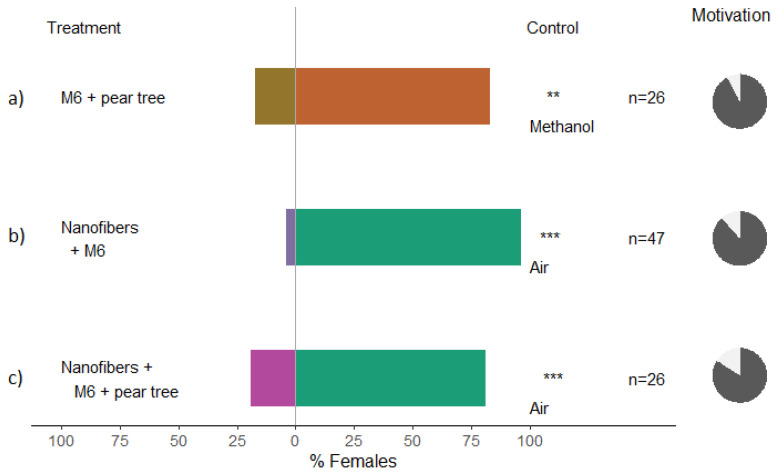
Results of Y-olfactometer-bioassays testing synthetic clove VOCs with adults of *Cacopsylla pyri* to establish whether they are repellent and could mask the odors of the host *Pyrus communis* cv. Williams Christ plants. (**a**) M6 (5% concentration in methanol) and a potted pear tree against methanol as control; (**b**) Nanofibers formulated with M6 (5% *v*/*v*) against air; (**c**) Nanofibers formulated with M6 (5% *v*/*v*) and a potted pear tree against air. The letter “n” corresponds to the numbers of insects used for the tests. Results are shown as the percentage of psyllids in each arm. Statistically significant differences are indicated with ** (*p* < 0.01); *** (*p* < 0.001). Percentage of psyllids that made a choice (dark gray) or not (light gray) is presented as pie charts on the right.

**Figure 5 insects-13-00743-f005:**
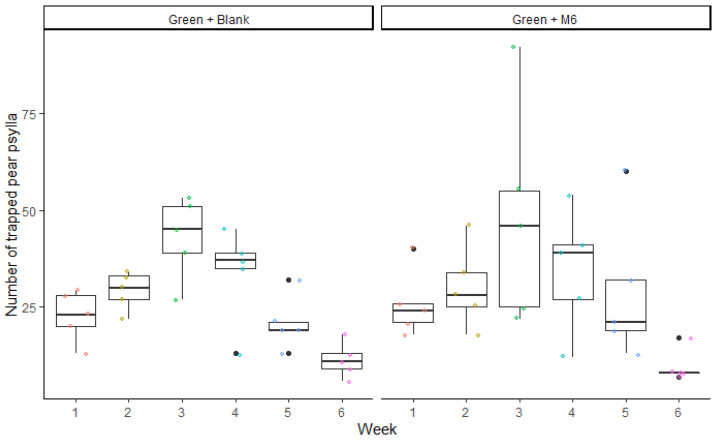
Average numbers of *Cacopsylla pyri* captured by five sticky traps in field experiment per week. The two trap types (green traps with blank nanofibers or green traps with M6-loaded nanofibers) were deployed in five blocks of pear trees (randomized block design) between 21 June 2021 and 2 August 2021. Boxes correspond to the 25th and 75th percentiles, medians appear as lines, and whiskers extend to 1.5 times of the interquartile ranges. Dots represent raw values.

**Figure 6 insects-13-00743-f006:**
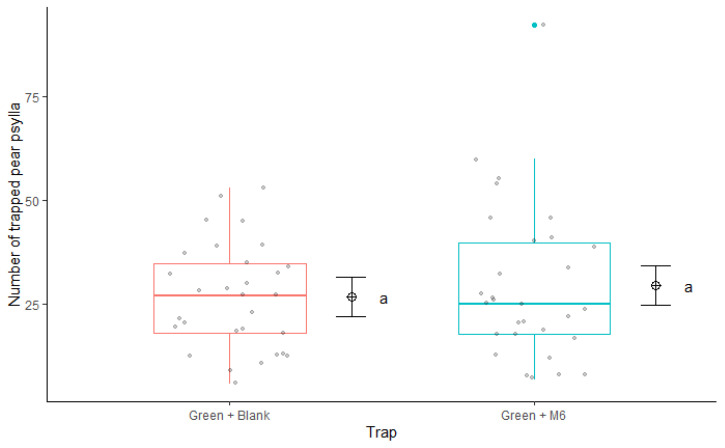
Statistical analysis of *Cacopsylla pyri* capture rates (time average from Figure 5) per trap-type (green traps equipped with blank nanofibers or green traps equipped with M6-loaded nanofibers) in field experiment using the Tukey test. The same letters (generalized linear model, Tukey test, *p* < 0.05) indicate no significant differences. Boxes correspond to the 25th and 75th percentiles, medians appear as lines, and whiskers extend to 1.5 times of the interquartile ranges. Dots represent raw values. Dots with crosses represent the means.

**Figure 7 insects-13-00743-f007:**
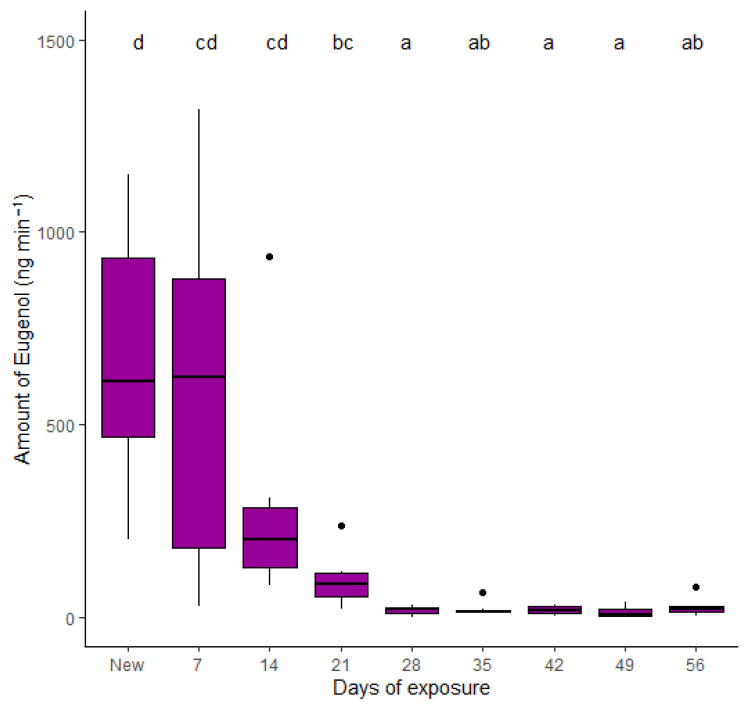
Release rates of eugenol (ng min^−1^) of PCL/CA nanofibers as a function of length of exposure over a period of 56 days. Measurements were performed once a week (n = 6). Different letters (generalized linear model, Tukey test, *p* < 0.05) indicate significant differences. Boxes correspond to the 25th and 75th percentiles, medians appear as lines, and whiskers extend to 1.5 times of the interquartile ranges. Dots represent outliers.

**Table 1 insects-13-00743-t001:** Components of the synthetic volatile blends comprising of the clove essential oil compounds used in olfactometer tests for pear psyllids (*Cacopsylla pyri* and *C. pyricola*).

Compounds	Amount (%)	M1	M2	M3	M4	M5	M6
Eugenol	88.6	+			+	+	+
Eugenyl acetate	8.9		+			+	+
β-Caryophyllene	1.9			+	+		+

## Data Availability

Data is available through the authors.

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
