# Peer review of "Repellent Activity of Clove Essential Oil Volatiles and Development of Nanofiber-Based Dispensers against Pear Psyllids (Hemiptera: Psyllidae)"

_insects, 2022, doi:10.3390/insects13080743_

Round 1

Reviewer 1 Report

This is an excellent manuscript and investigation. It is very well written and represented. The methods are well described and sound. The statistical analyses make perfect sense. The results are clear. The conclusions are well supported by the results. I only have minor suggestions stated below:

Line 17: Replace ‘besides’ with “Also” or “In addition”

Line 17: “nanofibers based” sounds awkward in pleural. Perhaps, “nanofiber-based”

Line 20: Volatiles “were” formulated change to pleural form of the verb.

Line 50: Italicize species name.

Line 207: I suggest briefly stating what the objective of this field experiment was or the hypothesis tested. I’m assuming it was to see if the repellent would reduce catch of psyllids as compared with the blank traps. It would be nice to specifically state this.

Line 236: Insert space at end of sentence.

Line 298, 300, 303, 304, 318, 321, 336, 340, 342, 343, 344, 353: Italicize species name.               

Figure 7. For the X-axis label, I suggest changing to “Days of exposure”         

Line 380: I suggest using the second “essential” and just say “close oil” in this sentence so as not to repeat the word; it’s understood.

Line 411-412: It would be nice to conduct an experiment where a larger area of crop is treated with the repellent formulation rather than a trap point source. Also, it would be nice to vary the application rate. Perhaps if a larger area were treated, the outcome would be different. Although this may have been beyond the scope of this current investigation, I think the authors may want to suggest these types of experiments for future efforts. I notice that this is very briefly mentioned at the end of the manuscript. I wonder if it would be appropriate to state something here regarding treating larger areas of crop with higher rates in order to explain the discrepancy between the laboratory and field outcomes.

Author Response

Dear reviewer,

Thank you for your pertinent and helpful comments.

The similarity found in some points of the materials and methods is due to the fact that this research work was carried out in the same laboratory and in the same period as the authors in Gallinger et al. 2020. Credit to the authors for publishing the methods was correctly given in the text of the manuscript with the citation of the work. I made some changes to the text to reduce the similarities found, but given the specificity of the description of the methods used, major changes are not possible.

All points suggested for review were taken into account and the modification was carried out as specified below:

Reviewer: Line 17: Replace ‘besides’ with “Also” or “In addition”

Authors: in Line 17 "besides" was replaced for " in addition"

Reviewer: Line 17: “nanofibers based” sounds awkward in pleural. Perhaps, “nanofiber-based”

Authors: in Line 17 “nanofibers based” was replace for “nanofiber-based”

Reviewer: Line 20: Volatiles “were” formulated change to pleural form of the verb.

Authors: in Line 20: the plural form of the verb "was" substituted for "were" 

Reviewer: Line 207: I suggest briefly stating what the objective of this field experiment was or the hypothesis tested. I’m assuming it was to see if the repellent would reduce catch of psyllids as compared with the blank traps. It would be nice to specifically state this.

Author: in line 211 the sentencewas added "Field experiments were conducted to observe if attractive stick traps equipped with nanofiber formulated with the repellent mixture (M6) would reduce the catches of psyllids in comparison with traps equipped with blank nanofiber. "

Reviewer: Line 236: Insert space at end of sentence.

Author: space inserted in line 236

Reviewer: Line 50, 298, 300, 303, 304, 318, 321, 336, 340, 342, 343, 344, 353: Italicize species name.  

Author: All species names were italicized          

Reviewer: Figure 7. For the X-axis label, I suggest changing to “Days of exposure”         

Authors:  In Figure 7.  X-axis label was changed to “Days of exposure”         

Reviewer: Line 380: I suggest using the second “essential” and just say “close oil” in this sentence so as not to repeat the word; it’s understood.

Authors: in line 380 we substitute the second "essential" for clove oil as in the following: "Like many other essential oils, clove oil is widely used as herbal medicine..."

Reviewer: Line 411-412: It would be nice to conduct an experiment where a larger area of crop is treated with the repellent formulation rather than a trap point source. Also, it would be nice to vary the application rate. Perhaps if a larger area were treated, the outcome would be different. Although this may have been beyond the scope of this current investigation, I think the authors may want to suggest these types of experiments for future efforts. I notice that this is very briefly mentioned at the end of the manuscript. I wonder if it would be appropriate to state something here regarding treating larger areas of crop with higher rates in order to explain the discrepancy between the laboratory and field outcomes.

Authors: We have added the following phrases to outline future research scopes: "This result can indicate that the cause of the treatment failure may be attributable to the distribution of the dispensers in the area rather than an ineffective repellent formulation. Perhaps the outcomes of the field trial could be different if a larger area of the orchard were treated with the repellent formulation rather than a trap point source. In this scenario, a substantial inhibitory effect on the pear psyllids population would be expected with a significant reduction in psyllids trapping. Such modifications in the experiment design could be performed for future investigations."

Reviewer 2 Report

The paper focuses on clove essential oil components to be used as repellents against pear psillids, and particularly on deploying the volatiles using nanofiber-based dispensers.

The work was conducted and analysed properly, and the results are very interesting and promising for the development of more sustainable pest management strategies using repellents.

Some parts in the methodology need better clarification.

Important: please check that all scientific names are correctly written in Italics throughout the manuscript.

Specific comments as follows.

L20 were instead of was

L24 nanofibers as volatile

L37-38 unclear, please rephrase to make it clearer

L51 spread instead of spreading

L53 scientific name in italics

L54 ..has been identified as the main vector

L108-9 The main component of clove essential oil… In any case I think that this paragraph should be better integrated with the one starting on L96

L168 room temperature and humidity

L183-4 it would be better to split the description of the experiments in different paragraphs, indicated with different numbers or letters

L184 and related paragraph and L192 Table 1: please specify the proportion of the substances in the mixtures (was it half-half for the 2 compounds mixtures and 1/3-1/3-1/3 for the 3 compound mixture?)

L184 and all paragraphs on the description of the experiment: please clearly indicate the total number of replicates performed for each trial (at least the range of the number of insects tested)

L200 I guess there is a mistake/something is missing here. “…contained the pear volatiles (M6) additionally…” but M6 was the acronym for the mixture with 3 compounds. Maybe the authors meant “pear volatiles + M6”? Please fix this.

L236 insert a space before “after”

L298, 300, 303, 304, 318, 321, 336….and more: scientific names in italics

Fig. 2, Fig. 3, Fig. 4, Fig. 5, Fig. 6 captions: scientific names in italics

L320 use a ; instead of the bracket here

L367 remove dot after 7

L400 add also the name of the other species, since tests were performed on both

L400-402 not clear, check this sentence and rephrase

L404 The with lower case letter for t

L422 a dot is missing between Fig and 7

References

Please check that all scientific names are correctly written in Italics, and also that all names of journals are written correctly

Author Response

Dear reviewer,

Thank you for your pertinent and helpful comments.

All points suggested for review were taken into account and the modification was carried out as specified below:

Reviewer: L20 were instead of was

Authors: in Line 20: the plural form of the verb "was" substituted for "were" 

Reviewer: L24 nanofibers as volatile

Authors: we changes the sentence for: "nanofibers as volatile dispensers"

Reviewer: L37-38 unclear, please rephrase to make it clearer

Authors: sentence rephrased: "In a pear orchard, we compared the captures of pear psyllids in green-colored attractive traps treated with nanofibers loaded with M6 mixture or unloaded nanofibers (blank)."

Reviewer: L51 spread instead of spreading

Authors: "Spreading" was changed to "spread"

Reviewer: L54 ..has been identified as the main vector

Authors: L54 found changed to identified

Reviewers: L108-9 The main component of clove essential oil… In any case I think that this paragraph should be better integrated with the one starting on L96

Authors: the order of the paragraphs was changed: "The major component of clove essential..." belongs now to the paragraph starting in L96

Reviewer: L168 room temperature and humidity

Authors: "humidity" added

Reviewer: L183-4 it would be better to split the description of the experiments in different paragraphs, indicated with different numbers or letters

Authors: Experiments were separated into different paragraphs and they were indicated with letters.

Reviewer: L184 and related paragraph and L192 Table 1: please specify the proportion of the substances in the mixtures (was it half-half for the 2 compounds mixtures and 1/3-1/3-1/3 for the 3 compound mixture?)

Authors: the proportion of each substance used in each mixture is indicated in Table 1 (amount %) as indicated in the text.

Reviewer: L184 and all paragraphs on the description of the experiment: please clearly indicate the total number of replicates performed for each trial (at least the range of the number of insects tested)

Authors: The total number of insects tested in each trial is indicated in the results section in the figures indicated as "n". The number of replicates varied in assay depending on the number of insects available.  We added a sentence in L183: " A minimum of 26 insects were tested in each trial."

Reviewer: L200 I guess there is a mistake/something is missing here. “…contained the pear volatiles (M6) additionally…” but M6 was the acronym for the mixture with 3 compounds. Maybe the authors meant “pear volatiles + M6”? Please fix this.

Authors: in L202 the following modification in the sentence were done: "In the olfactometer arm contained the pear volatiles, a filter paper containing the M6 mixture, as described above, was added, and the other arm was supplied only with solvent."

Reviewer: L236 insert a space before “after”

Authors: space added

Reviewer: L53, L298, 300, 303, 304, 318, 321, 336….and more: scientific names in italics

Authors: All species names were italicized           

Reviewer: Fig. 2, Fig. 3, Fig. 4, Fig. 5, Fig. 6 captions: scientific names in italics

Authors: All species names were italicized as suggested

Reviewer: L320 use a ; instead of the bracket here

Authors: The brackets were removed in the following: "(significant p-values from binomial tests; Fig. 4a, c)"

Reviewer: L367 remove dot after 7

Authors: changed "Fig. 7"

Reviewer: L400 add also the name of the other species, since tests were performed on both

Authors: We changed the sentence for better understanding: "By conducting choice assays, we evaluated the relative preference or avoidance between single compounds and mixtures of clove essential oils for their ability to repel C. pyri and C. pyricola, and in altering host-seeking behavior in C. pyri."

Reviewer: L400-402 not clear, check this sentence and rephrase

Authors: Sentence was modified to: "Experiments with this configuration which assess the number of insects avoiding treated plants in the olfactometer may be more predictive of insect's behavior in the field."

Reviewer: L404 The with lower case letter for t

Authors: lower case changed

Reviewer: L422 a dot is missing between Fig and 7

Authors: point added

Reviewer: References

Please check that all scientific names are correctly written in Italics, and also that all names of journals are written correctly

Authors: all references were checked and doi numbers were added.